# Low-FODMAP Diet for the Management of Irritable Bowel Syndrome in Remission of IBD

**DOI:** 10.3390/nu14214562

**Published:** 2022-10-29

**Authors:** Martyna Więcek, Paulina Panufnik, Magdalena Kaniewska, Konrad Lewandowski, Grażyna Rydzewska

**Affiliations:** 1Clinical Department of Internal Medicine and Gastroenterology with Inflammatory Bowel Disease Subunit, Central Clinical Hospital of Ministry of the Interior and Administration, 02-507 Warsaw, Poland; 2Collegium Medicum, Jan Kochanowski University, 25-369 Kielce, Poland

**Keywords:** irritable bowel syndrome, inflammatory bowel disease, low-FODMAP diet

## Abstract

Approximately 30% of patients with quiescent inflammatory bowel disease (IBD) meet the diagnostic criteria for irritable bowel syndrome (IBS). The aim of this study was to evaluate the effectiveness of a low-FODMAP diet in patients who meet the diagnostic criteria for IBS whilst in IBD remission. A total of 200 patients in remission of IBD were included in the study. Sixty-five of these patients (32.5%) were diagnosed with IBS according to the R4DQ. On the patients who met the IBS diagnostic criteria, anthropometric measurements, laboratory tests and lactulose hydrogen breath tests were performed. A low-FODMAP diet was introduced for 6 weeks. Of the 59 patients with IBS diagnosed at baseline for whom data were collected at the end of follow-up, after the low-FODMAP intervention IBS-like symptoms were not present in 66.1% (*n* = 39) (95% CI (53.4%; 76.9%)). The difference between the two groups (with SIBO at baseline (33 of 48 patients) and without SIBO at baseline (6 of 11 patients)) in the low-FODMAP diet’s effectiveness was not statistically significant (*p* = 0.586). The low-FODMAP diet improved the gut symptoms of flatulence and diarrhea. It had no effect on the occurrence of constipation. In IBD patients in remission who meet the IBS criteria, the dietary intervention of a low-FODMAP diet is effective for a reduction in IBS-like symptoms, regardless of the coexistence of bacterial overgrowth.

## 1. Introduction

Crohn’s disease (CD) and ulcerative colitis (UC) are types of inflammatory bowel disease (IBD) with complex and complicated pathogenesis. Environmental factors as well as genetic predisposition play a substantial role in their development. Many recent studies have shown several alterations in the gut microbiota in IBD patients [1,2], manifesting as an overall drop in species diversity. This is primarily a consequence of fewer *Firmicutes* spp. (e.g., the genus *Lactobacillus*), *Bacteroides* spp. and *Bifidobacterium* spp. [3]. On the other hand, species which are more numerous in IBD are *Eschericha coli*, *Fuscobacterium*, *Haemophilus parainfluenzae*, *Eikenella corrodens* and *Gemella moribillum* [4]. Moreover, in IBD the intestinal mucosa is more permeable for bacterial antigens, leading to the immune system being activated. Proinflammatory factors are released and negatively affect the epithelial function [5,6]. 

The natural course of IBD involves periods of active disease interspersed with periods of remission. From time to time, patients experience gastrointestinal symptoms which, although they do not result from activation of the underlying disease, are a manifestation of functional disorders, such as irritable bowel syndrome (IBS). IBS, according to Rome IV diagnostic criteria, is a combination of recurrent abdominal pain associated with a change in form or frequency of stool [7]. Studies highlight the presence of mucosal inflammation at the microscopic and molecular level in IBS [8]. On the other side, disturbance to the intestinal microbiota is reported [9]. Systematic reviews and meta-analyses suggest a link between IBS and small intestinal bacterial overgrowth (SIBO). Depending on the diagnostic method, the prevalence of SIBO in IBS varies from 14% to 40% [10,11]. The symptoms of SIBO are a result of disturbances in the mechanisms responsible for maintaining normal levels of bacterial colonization in the lumen of the small intestine [12]. Among many other factors, IBD predisposes the patient to imbalanced bacterial homeostasis and therefore the development of SIBO [13]. Treatment using broad-spectrum antibiotics, including rifaximin, is recommended [12]. Apart from pharmacological intervention, a low-FODMAP diet can be useful in alleviating the symptoms of SIBO and IBS [14].

A low-FODMAP dietary intervention calls for the restriction of monosaccharides (e.g., fructose), disaccharides (e.g., lactose), oligosaccharides (e.g., fructans and galactans) and polyols (e.g., sorbitol and xylitol) in the patient’s everyday diet. This leads to fewer readily fermentable substrates in the distal small intestine and proximal colon, which are responsible for luminal distension and functional gut symptoms [15]. 

The use of a low-FODMAP diet among patients with quiescent IBD has significant clinical implications. It allows the avoidance of an excessive use of steroids or premature intensification of the IBD treatment in patients with abdominal symptoms without increased inflammatory parameters. At the same time, the incidence of opportunistic diseases such as melanomas, tuberculosis, hepatitis B virus, hepatitis C virus, human immunodeficiency virus infections and steroid-dependent complications can be reduced.

Nevertheless, clinicians must be aware of possible side effects of an elimination diet. Low-FODMAP intervention is not meant for long-term use as it can lead to malnutrition, which can manifest as iron-deficiency anemia, osteoporosis, impaired tissue healing or neuropathy [16]. This translates to poor clinical outcomes, response to therapy and, therefore, quality of life [17].

## 2. Materials and Methods

Between January and April 2022, 200 patients from a gastroenterology outpatient center in Central Clinical Hospital of the Ministry of Interior and Administration in Warsaw, Poland with CD or UC who were in remission of IBD were examined using the Rome IV Diagnostic Questionnaire (R4DQ) for Adult FGIDs—Irritable Bowel Syndrome Module in order to diagnose IBS (Figure 1). Subjects with diagnosed IBS were admitted for a one-day hospitalization in the Clinical Department of Internal Medicine and Gastroenterology with the Inflammatory Bowel Disease Subunit, Central Clinical Hospital of the Ministry of Interior and Administration in Warsaw, Poland to undergo all tests included in the study protocol in week 0 and week 6 of the study. 

The primary endpoint of the study was to evaluate the effectiveness of a low-FODMAP diet in patients with IBD in clinical remission who met the criteria for IBS. The secondary endpoints were evaluating the prevalence of IBS and SIBO in IBD-remission patients, comparing the effectiveness of a low-FODMAP diet in IBS patients with and without SIBO and assessing the impact of a low-FODMAP diet on nutritional, hepatic, renal and inflammatory parameters. 

CD patients with a calculated Crohn’s disease activity index (CDAI) value of <150 points or UC patients with a partial Mayo score (PMS) of <2 points and a fecal calprotectin level of <250 µg/g were included in the study. 

The exclusion criteria for the study were: age > 80 years, a lack of cooperation with the researcher, being in the active phase of the disease (CDAI ≥ 150 points, PMS ≥ 2 points), a fecal calprotectin level of ≥250 µg/g, changes in treatment (medications, supplements) in the 8 weeks prior to qualification and/or during the study, positive GDH and/or stool cultures at the time of qualification and/or during the study, any dietary interventions 4 weeks prior to qualification, antibiotic use 4 weeks prior to qualification and/or during the study, the use of probiotics, prokinetics, antidiarrheal drugs, proton-pump inhibitors or H2-receptor blockers 7 days prior to qualification and/or during the study, pregnancy or breastfeeding, lactulose intolerance and a diagnosis of galactosemia. 

The patients who met the IBS diagnostic criteria were seen at baseline and at week 6 of the study. During these visits, anthropometric measurements (height, weight and BMI) were taken, CDAI and PMS were calculated, laboratory tests (peripheral blood count, CRP, ESR, creatinine, urea, uric acid, ALT, AST, total bilirubin, ALP, GGTP, albumin, total protein, sodium, potassium, calcium, iron, ferritin, vitamin B12, folic acid, vitamin D3 and calprotectin levels and a fecal culture) and a lactulose hydrogen breath test (LHBT) were performed. After consumption of 10 g (20 mL) lactulose, the rise in breath hydrogen by 20 ppm above the basal within 90 min after ingestion of lactulose was considered as diagnostic of SIBO. The severity of the symptoms (flatulence, diarrhea, constipation) was determined subjectively by patients at baseline and at week 6 of the study. Low-FODMAP effectiveness was defined as complete resolution of abdominal pain and diminished symptoms of diarrhea and/or constipation. It was assessed subjectively by the patients and reported to the physician during the follow-up interview.

In patients with a diagnosis of IBS, a low-FODMAP dietary intervention was introduced regardless of the LHBT result. During the dietary consultation, the patients received detailed written instructions regarding the diet with a list of allowed and disallowed food products as well as sample recipes. The excluded products with excessive amounts of FODMAPs were fruits and vegetables like apples, pears, peaches, plums, mangoes, watermelon, broccoli, cauliflower, garlic, onions and peas. The suitable alternatives were bananas, blueberries, grapes, lemons, limes, oranges, raspberries, strawberries, tomatoes, carrots, corn, eggplants and lettuce. Honey had to be replaced with maple syrup or any other sweetener except polyols (sorbitol, mannitol, xylitol). The consumption of cow’s, goat’s and sheep’s milk was not allowed—patients had to choose lactose-free substitutes. Wheat and rye in any form (in bread, pasta, biscuits) was excluded [15]. 

Patients were advised to keep food diaries—a 7-day food diary written down by participants a week prior to the follow-up visit. They were supposed to be completed immediately after meal consumption. According to the study protocol, at least 90% of the patients’ meals had to fulfil the low-FODMAP diet recommendations in order to be considered good compliance. The diaries were presented to a dietician on follow-up visit. On the follow-up consultation with a dietician, the patients were advised to reintroduce eliminated food products, one after another, with careful observation of the body’s reaction to the introduced product. This was to adjust the patients’ dietary plans to minimize the occurrence of gastrointestinal symptoms.

Statistical analysis was carried out using the software program R, version 4.0.5. Continuous variables were presented as mean ± SD or median (Q1; Q3 (lower and upper quartiles, respectively)). Nominal variables were presented as count n (% frequency). The Shapiro–Wilk test, the skewness and kurtosis values and a visual assessment of histograms were used to validate the normality of the distribution. The two measurements (baseline and follow-up) were compared with a paired t-test or Wilcoxon signed-rank test, and the mean/median differences (MD) with 95% confidence intervals (CI) were calculated. McNemar’s test was used to compare nominal variables between baseline and follow-up, while the chi-square test was used to compare the effectiveness of a low-FODMAP diet between groups. To compare the effectiveness of a low-FODMAP diet between IBS-C, IBS-D and IBS-U subgroups, Fisher’s exact test was used; 95% binomial confidence intervals (CI) were calculated for proportions where relevant. The tests were based on α = 0.05.

## 3. Results

A total of 200 patients in remission of IBD were included in the study. Sixty-five of these patients (32.5%) were diagnosed with IBS according to the R4DQ (Figure 2a). Moreover, 7 patients had IBS with predominant constipation (IBS-C), 26 had IBS with predominant diarrhea (IBS-D), 1 patient suffered from IBS with mixed bowel habits (IBS-M) and 31 participants were diagnosed with unclassified IBS (IBS-U). Of the IBS group, 44.6% (*n* = 29) suffered from UC and 55.4% (*n* = 36) suffered from CD. Six patients initially consented to participate in the study but did not appear at the follow-up appointment. The baseline characteristics of the IBS patients are presented in Table 1. 

Of the group of patients in remission of IBD who met the diagnostic criteria for IBS, SIBO was diagnosed in 80% (*n* = 52) (Figure 2b). 

The effect of a 6-week low-FODMAP dietary intervention on laboratory test parameters is presented in Table 2. 

At baseline in the IBS group, 25 patients (42.4%) reported diarrhea, while 9 patients (15.3%) did so at the end of follow-up (*p* < 0.001) (Figure 3a). Constipation was reported at baseline by 8 patients (13.6%) in the IBS group, compared with 5 patients (8.5%) after the dietary intervention (*p* = 0.453) (Figure 3b). Flatulence was reported by 94.9% (*n* = 56 patients) of the IBS group at baseline, while at the end of follow-up it affected 27.1% (*n* = 16 patients) (*p* < 0.001) (Figure 3c). 

Among the 59 patients diagnosed with IBS at baseline for whom data were collected at the end of follow-up, after the 6-week low-FODMAP intervention IBS-like symptoms were not present in 66.1% (*n* = 39) (95% CI (53.4%; 76.9%)) (Figure 4a). The effectiveness of a low-FODMAP diet did not differ between ulcerative colitis (59.3%) and Crohn’s disease (71.9%) patients (*p* = 0.457).

The effectiveness of a low-FODMAP diet in patients with predominant constipation (IBS-C subgroup) was 71.4% (95% CI (29.1%; 96.3%)) (5 of 7 patients), in patients with predominant diarrhea (IBS-D) it was 66.7% (95% CI (44.7%; 84.4%)) (16 of 24 patients), in patients with unclassified IBS (IBS-U) it was 63.0% (95% CI (42.4%; 80.6%) (17 of 27 patients). No statistically significant difference in low-FODMAP effectiveness was observed between the IBS-C, IBS-D and IBS-U subgroups (*p* > 0.999).

In the SIBO subgroup of 48 subjects—with both IBS and SIBO at baseline—68.8% (*n* = 33) (95% CI (53.8%; 81.3%)) did not experience IBS-like symptoms at the end of follow-up (Figure 4b). In the subgroup of 11 IBS patients without SIBO at baseline for whom data were collected at the end of follow-up, 54.5% (*n* = 6) (95% CI (28.0%; 78.7%)) did not report IBS-like symptoms (Figure 4c). The difference in the effectiveness of the low-FODMAP diet between the two groups (with SIBO at baseline (33 of 48 patients) and without SIBO at baseline (6 of 11 patients)) was not statistically significant (*p* = 0.586).

The diet adherence at the end of the 6-week program was 96.6% (*n* = 57 of 59 patients).

For the comparison of low-FODMAP effectiveness between the two groups (IBS and SIBO at baseline and IBS without SIBO at baseline), a post hoc power calculation was made using G*Power 3.1.9.2. Assuming α = 0.05, a power (1-β) of 67.8% was achieved.

## 4. Discussion

The prevalence of symptoms meeting the criteria for IBS in patients in remission of IBD found in our study is 32.5% and corresponds to the results of meta-analyses from 2020 and 2012, in which IBS was reported in 32.5–35.0% of the patients with quiescent IBD [18,19,20]. SIBO was diagnosed in 80% of our IBS participants. This was much higher than the average proportion of SIBO in IBS in the general population, which varies from 14% to 40%, depending on the diagnostic method [8,10,21,22]. In our research, the low-FODMAP dietary intervention was effective regardless of the coexistence of SIBO. The study proved that a low-FODMAP diet could be an available therapeutic option in the treatment of IBS in quiescent IBD.

Our 6-week low-FODMAP dietary intervention had no impact on IBD activity markers of inflammation (fecal calprotectin and CRP levels in blood serum). According to the literature, in a study on 55 Italian patients with IBD in remission or with a mild form of the disease, a short-term low-FODMAP diet was associated with an amelioration of inflammatory markers [23]. In our research, patients with an increased calprotectin level at week 0 were excluded from the study. We therefore did not observe any improvement in inflammatory parameters. Nevertheless, the intervention itself did not lead to an activation of IBD.

No changes were observed in total protein, iron, ferritin, vitamin B12 or folic acid concentrations. There was a statistically significant change in WBC, PLT, RBC, hemoglobin, hematocrit, ALP, sodium and albumin concentrations. The mean values of these parameters were still within the reference range and therefore not clinically significant. However, we observed a drop in BMI level on the follow-up visit. This indicates the need to monitor patients in order to prevent malnutrition whilst on a low-FODMAP diet. The data from the literature suggest potential nutritional deficiencies and an impact on carbohydrate, fiber and iron intake from a low-FODMAP diet, which is understandable considering the restrictive nature of this elimination diet [14].

In our study, the low-FODMAP diet reduced the severity of gut symptoms in IBD remission patients. As expected from the mechanism of action, the greatest impact was observed regarding flatulence. The incidence of diarrhea was also significantly lower after the 6-week intervention. It had no effect on the occurrence of constipation. The results are consistent with data from the literature on IBS [24,25]. The lack of an effect on constipation may be related to the reduced supply of fiber on a low-FODMAP diet, which has been reported in scientific studies [26]. However, the low number of patients reporting constipation at baseline might have influenced the analysis. Cox et al. in a randomized controlled trial reported a greater reduction in the IBS severity scoring system (IBS-SSS), reduced severity of flatulence and significantly lower daily stool frequency in IBD patients in remission on a low-FODMAP diet vs. participants on a sham diet. However, low-FODMAP had no effect on abdominal pain or stool consistency [27]. In a review from 2022 which included nine trials (a total of 446 participants with active and quiescent IBD), it was stated that a low-FODMAP diet improved functional gastrointestinal symptoms and quality of life but did not affect stool consistency or mucosal inflammation [28].

In our study, the effectiveness of the low-FODMAP intervention in the IBS group with and without SIBO did not differ significantly. This indicates that in IBS patients a low-FODMAP diet is effective regardless of the coexistence of bacterial overgrowth and can therefore be recommended for any IBS patient with low inflammatory parameters. This finding is important as it eliminates the need to perform a time-consuming and sometimes unavailable LHBT for the diagnosis of SIBO. Hydrogen breath tests are often misused by many clinicians despite the high risk of false positive and false negative results. The results of the study should encourage clinicians to have a more thoughtful approach when referring patients to LHBT.

A major limitation of this single-center study is the lack of randomization and a control group and the before/after nature of the trial. However, the study design did include a control group for SIBO patients. To limit recall bias, patients had to fulfil food diaries. They were supposed to be completed immediately after meal consumption. Its self-reported nature might have influenced the compliance. The severity of the symptoms was determined subjectively by patients. We did not use any objective questionnaires, which can affect the obtained results. To exclude the potential effect of medications and supplements on IBS symptoms, no changes to the treatment were allowed in the 8 weeks prior to qualification and/or during the study. Even supplements, such as vitamin D, in clinical studies improved IBS symptom severity scores and increased the quality of life [29].

The compliance in our study was 96.6%. Six patients did not appear at the follow-up appointment, which indicates that the actual adherence was lower. The compliance to a low-FODMAP diet has been reported in the IBS in literature as 40–80% [30,31]. In a trial by Cox et al. that concerned IBD patients, it was 88% [27]. Several factors could have led to the high adherence reported in our study. According to the protocol, 10% of the meals—which is an average of one meal every 2 days—did not have to be prepared according to the low-FODMAP recommendations; this loose assumption might have affected the results. The compliance in the study may have been high because of the relatively small group of participants who received much more attention than in the standard health care system. Moreover, IBD patients suffer from a chronic disease with potentially serious complications. They therefore might have been motivated to follow the proposed guidelines.

## 5. Conclusions

The role of environmental factors in the pathogenesis of IBS explains the beneficial effects of dietary interventions in clinical practice. The study provides evidence that a low-FODMAP dietary intervention is effective for all patients with IBS-like symptoms, regardless of the coexistence of SIBO. It may encourage clinicians to introduce a low-FODMAP diet without an LHBT. Subsequent studies on a low-FODMAP diet may provide further evidence of its effectiveness in functional and inflammatory gastrointestinal disorders.

## Figures and Tables

**Figure 1 nutrients-14-04562-f001:**
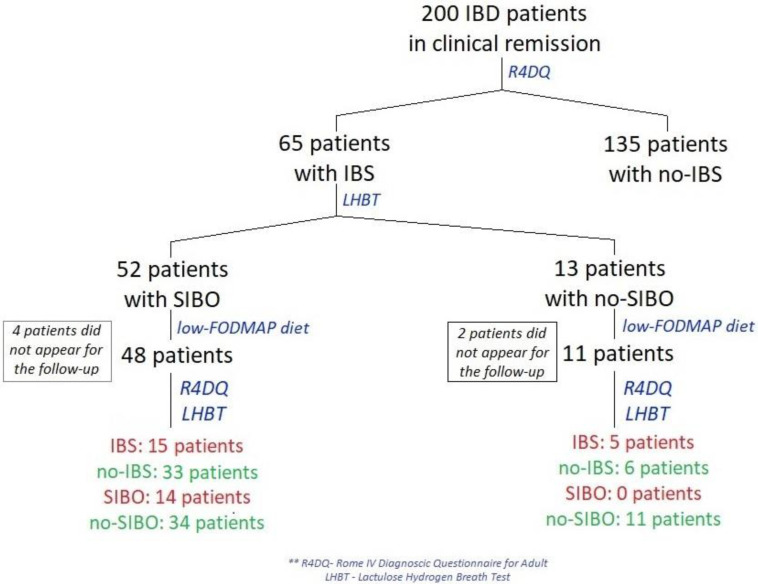
Flow diagram of patients in remission of IBD treated with a low-FODMAP diet.

**Figure 2 nutrients-14-04562-f002:**
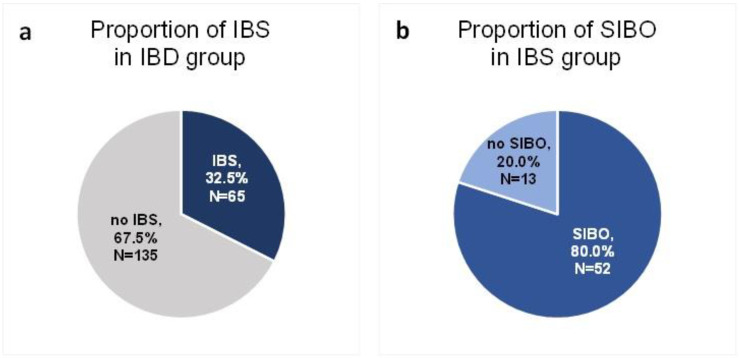
Proportion of IBS (**a**) and SIBO (**b**) in the study group.

**Figure 3 nutrients-14-04562-f003:**
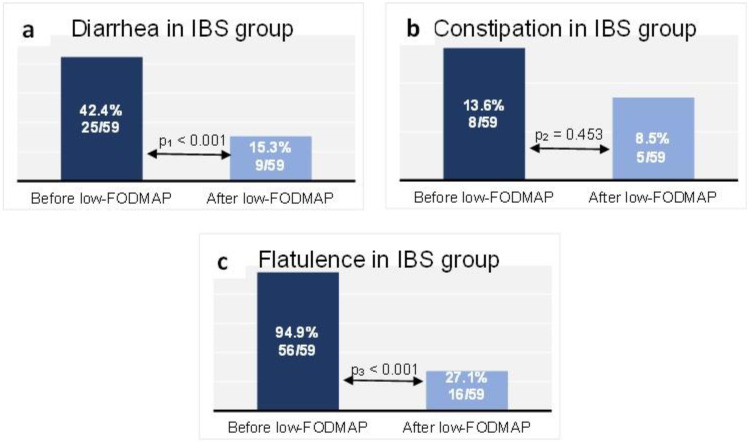
Proportion of the IBS group reporting diarrhea (**a**), constipation (**b**) and flatulence (**c**) at baseline and at the end of follow-up. Data are presented as n (% of total group). McNemar test: *p*_1_ < 0.001, *p*_2_ = 0.453, *p*_3_ < 0.001.

**Figure 4 nutrients-14-04562-f004:**
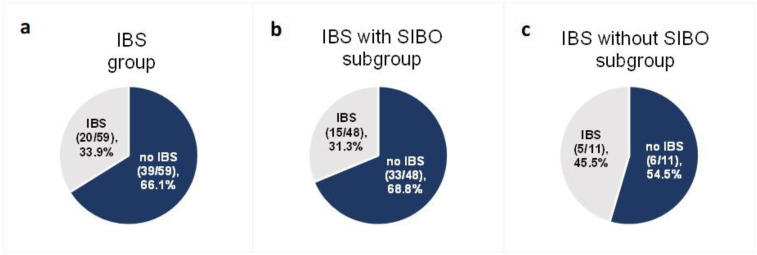
Proportion of patients with IBS-like symptoms after the low-FODMAP dietary intervention in the group with IBS at baseline (**a**) and in the two IBS subgroups: with SIBO at baseline (**b**) and without SIBO at baseline (**c**).

**Table 1 nutrients-14-04562-t001:** Baseline characteristics of IBS patients.

Variable	N	Level	Range
N	65	100.0%	
SIBO, n (%)	65	52 (80.0)	
Sex, female, n (%)	65	37 (56.9)	
Age at the beginning of the study, years, mean ± SD	65	43.33 ± 12.99	27–79
Weight, kg, mean ± SD	65	72.45 ± 15.58	41–107
Height, cm, mean ± SD	65	170.66 ± 9.13	153–190
BMI, mean ± SD	65	24.72 ± 4.14	15.05–34.90
Diarrhea, n (%)	65	27 (41.5)	
Constipation, n (%)	65	8 (12.3)	
Flatulence, n (%)	65	61 (93.8)	
Disease, n (%)			
UC	65	29 (44.6)	
CD		36 (55.4)	

**Table 2 nutrients-14-04562-t002:** Changes in the parameters of IBS patients.

Parameter	Baseline Visit at 0 Weeks	Follow-Up Visit at 6 Weeks	MD (95% CI)	*p*
Body weight, kg	72.07 ± 14.98	71.42 ± 14.55	−0.65 (1.28; −0.02)	0.043
BMI, kg/m^2^	24.49 ± 3.95	24.18 ± 3.85	−0.31 (−0.54; −0.08)	0.009
Calprotectin, µg/g	33.70 (30.00; 67.00)	32.00 (27.00; 71.00)	−3.00 (−12.85; 0.50)	0.077 ^1^
WBC, 10^3^/µL	6.73 ± 2.00	5.90 ± 1.97	−0.78 (−1.16; −0.41)	<0.001
RBC, 10^6^/µL	4.68 ± 0.47	4.65 ± 0.48	−0.06 (−0.12; −0.01)	0.019
PLT, 10^3^/µL	287.22 ± 78.27	272.28 ± 77.55	−12.82 (−21.16; −4.49)	0.003
Hgb, g/µL	14.03 ± 1.56	13.92 ± 1.62	−0.22 (−0.40; −0.04)	0.018
Ht, %	41.93 ± 3.64	41.33 ± 3.82	−0.79 (−1.33; −0.26)	0.004
ESR, mm/h	9.00 (4.00; 15.00)	8.00 (2.00; 51.00)	0.00 (−1.50; 1.00)	0.548 ^1^
CRP, mg/L	1.35 (0.50; 2.70)	1.10 (0.20; 25.20)	0.00 (−0.70; 0.10)	0.129 ^1^
AlAT, U/L	17.50 (12.75; 27.25)	19.00 (10.00; 47.00)	0.00 (−2.00; 2.50)	0.793 ^1^
AspAT, U/L	19.00 (16.00; 23.25)	21.00 (12.00; 71.00)	1.00 (−0.50; 2.00)	0.238 ^1^
Bilirubin, mg/dL	0.53 (0.40; 0.74)	0.54 (0.00; 1.16)	−0.03 (−0.08; 0.01)	0.171 ^1^
ALP, U/L	76.19 ± 21.09	72.93 ± 18.65	−2.47 (−4.67; −0.27)	0.029
GGTP, U/L	19.00 (13.25; 28.75)	16.00 (12.50; 26.50)	−1.00 (−3.00; 0.50)	0.136 ^1^
Creatinine, mg/dL	0.79 (0.71; 0.90)	0.78 (0.69; 0.90)	0.00 (−0.03; 0.04)	0.726 ^1^
Urea, mg/dL	27.76 ± 7.88	27.73 ± 6.48	−0.81 (−2.27; 0.65)	0.269
Uric acid, mg/dL	4.50 (3.90; 5.33)	4.50 (3.90; 5.25)	0.10 (−0.01; 0.40)	0.073 ^1^
Na, mmol/L	141.23 ± 2.00	140.58 ± 2.06	−0.76 (−1.40; −0.12)	0.021
K, mmol/L	4.36 ± 0.39	4.39 ± 0.33	0.05 (−0.05; 0.14)	0.323
Ca, mmol/L	2.38 ± 0.11	2.37 ± 0.11	−0.02 (−0.04; 0.01)	0.171
Fe, µg/dL	106.25 ± 44.52	96.80 ± 45.12	−9.60 (−21.99; 2.81)	0.127
Ferritin, ng/mL	73.00 (32.00; 146.25)	71.00 (35.00; 148.50)	−5.00 (−10.00; 4.00)	0.285 ^1^
Vit. B12, pg/mL	396.89 ± 143.92	385.51 ± 136.60	−16.36 (−37.71; 4.99)	0.130
Folic acid, ng/mL	7.00 (5.20; 12.00)	7.50 (5.30; 11.55)	0.15 (−0.50; 0.85)	0.597 ^1^
Albumin, g/dL	4.68 (4.53; 4.79)	4.55 (4.43; 4.73)	−0.13 (−0.20; −0.08)	<0.001 ^1^
Total protein, g/dL	7.26 (7.02; 7.56)	7.22 (6.84; 7.44)	−0.19 (−0.28; 0.01)	0.065 ^1^
Vit. D3, ng/mL	26.30 (20.80; 35.50)	28.20 (21.70; 38.40)	0.25 (−0.65; 3.15)	0.206 ^1^

Data are presented as mean ± SD or median (Q1; Q3), depending on the normality of the distribution. MD—mean/median difference (follow-up minus baseline level) with 95% confidence interval (CI). ^1^ Both time-points were compared with a paired *t*-test or Wilcoxon test.

## Data Availability

Data supporting reported results are available upon request.

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
