# Peer review of "Low-FODMAP Diet for the Management of Irritable Bowel Syndrome in Remission of IBD"

_nutrients, 2022, doi:10.3390/nu14214562_

Round 1

Reviewer 1 Report

There are multiple methodological issues with the present study. Please refer below for my specific comments and suggestions.

Specific comments:

1. Suggest including the definition of IBS in the introduction section. It is a combination of chronic abdominal pain associated with a change in the frequency or form of stool.

2. "... a link between IBS and small intestinal bacterial overgrowth (SIBO)" - apart from this, chronic, low-grade, subclinical inflammation has been implicated in the disease process and is thought to perpetuate IBS-like symptoms (citation: ncbi.nlm.nih.gov/pmc/articles/PMC6159811).

3. It is unclear how the patients for the study were recruited. Were these patients being followed up at a tertiary hospital or specialist centre?

4. Was there an age cut-off for the study participants?

5. It is important to specify if the subjects were IBS-C, IBS-D or IBS-M. This has important implications for treatment.

6. What about subjects taking vitamin supplements? Vitamin D may have beneficial effects on IBS symptoms (citation: pubmed.ncbi.nlm.nih.gov/35396764). This should be acknowledged.

7. "The two measurements (baseline and follow-up) were compared with a paired t-test or Wilcoxon signed-rank test and the mean/median differences (MD) with 95% confidence intervals (CI) were calculated. McNemar’s test was used to compare nominal variables between baseline and follow-up, while the chi-square test was used to compare the effectiveness of a low-FODMAP diet between groups" - for a “repeated-measures” scenario (multiple intervals or considerable matching between groups), you need to reduce the alpha level using the Bonferroni correction method.

8. Please provide the actual institutional review board (IRB) study/approval number for reference.

9. Please change "CI95" to "95% CI".

10.  The conceptual basis for a low-FODMAP diet seems to be misunderstood by the authors. The low-FODMAP diet is a temporary learning diet designed to help ameliorate digestive symptoms. The foods are not absolute or universal for every patient with IBS; some patients do fine with oranges while others are triggered by it. It is an elimination diet that should only be done from 2-6 weeks. After which, you are encouraged to slowly reintroduce FODMAP-rich foods back into your diet. The idea behind the short-term elimination of FODMAP-rich foods from patients' diets is to figure out exactly which FODMAP-rich foods are causing digestive discomfort. Once patients have identified which particular FODMAP-rich foods are triggering their symptoms, they are then in a better position to adjust their dietary plans and minimise the occurrence of bothersome symptoms.

11. The discussion of study limitations was incomplete. What about recall issues/bias?

12. Please have a separate conclusions section.

13. Please include a data availability statement. The underlying data should be made publicly available. If this was not possible, please provide a reason why.

Author Response

Thank you for your comments and suggestions to the manuscript. We would be very grateful
to reply as detailed as possible.
1. Suggest including the definition of IBS in the introduction section. It is a combination of
chronic abdominal pain associated with a change in the frequency or form of stool.
We included the definition of IBS in the introduction section (lines 42-44) as suggested.
2. "... a link between IBS and small intestinal bacterial overgrowth (SIBO)" - apart from this,
chronic, low-grade, subclinical inflammation has been implicated in the disease process and
is thought to perpetuate IBS-like symptoms (citation:
ncbi.nlm.nih.gov/pmc/articles/PMC6159811).
We added that information in lines 44-45.
3. It is unclear how the patients for the study were recruited. Were these patients being
followed up at a tertiary hospital or specialist centre?
200 patients from a gastroenterology outpatient centre (Central Clinical Hospital of the
Ministry of Interior and Administration in Warsaw, Poland) were recruited for the study.
Those with diagnosed IBS were admitted for a one-day hospitalisation in Clinical Department
of Internal Medicine and Gastroenterology with Inflammatory Bowel Disease Subunit, Central
Clinical Hospital of Ministry of the Interior and Administration in Warsaw, Poland to perform
all tests included in the study protocol in week 0 and week 6 of the study. We updated that
information in lines 67-75.
4. Was there an age cut-off for the study participants?
The age cut-off for the study participants was 80 years old. The oldest participant was 79 years
old. We added that information in line 85.
5. It is important to specify if the subjects were IBS-C, IBS-D or IBS-M. This has important
implications for treatment.
We added thisinformation in lines 141-144. 7 patients had IBS with predominant constipation
(IBS-C), 26 had IBS with predominant diarrhoea (IBS-D), 1 patient suffered from IBS with mixed
bowel habits (IBS-M), 31 participants were diagnosed with unclassified IBS (IBS-U).
6. What about subjects taking vitamin supplements? Vitamin D may have beneficial effects
on IBS symptoms (citation: pubmed.ncbi.nlm.nih.gov/35396764). This should be
acknowledged.
We did not collect information on vitamin D supplementation. The patients enrolled in the
study had no modifications in the use of medications and supplements during the last 8 weeks
before the study to avoid its potential impact on the effect of the low-FODMAP diet. We
specified that in lines 87-88.
7. "The two measurements (baseline and follow-up) were compared with a paired t-test or
Wilcoxon signed-rank test and the mean/median differences (MD) with 95% confidence
intervals (CI) were calculated. McNemar’s test was used to compare nominal variables
between baseline and follow-up, while the chi-square test was used to compare the
effectiveness of a low-FODMAP diet between groups" - for a “repeated-measures” scenario
(multiple intervals or considerable matching between groups), you need to reduce the
alpha level using the Bonferroni correction method.
Thank you for this comment. We however believe that Bonferroni correction is not applicable
in this case. Bonferroni correction is used for multiple comparisons of the same group (i.e.
when there are at least 2 comparisons). In our case if we had data from at least 3 time points,
Bonferroni correction would be a good approach (there would be 3 comparisons: 1 vs. 2, 1 vs.
3, 2 vs. 3). In our case we have only 2 time points, so there is only one comparison (baseline
vs. follow-up), so Bonferroni correction for multiple comparisons is not applicable.
8. Please provide the actual institutional review board (IRB) study/approval number for
reference.
The information is provided in lines 349-352.
9. Please change "CI95" to "95% CI".
We changed that as requested.
10. The conceptual basis for a low-FODMAP diet seems to be misunderstood by the
authors. The low-FODMAP diet is a temporary learning diet designed to help ameliorate
digestive symptoms. The foods are not absolute or universal for every patient with IBS;
some patients do fine with oranges while others are triggered by it. It is an elimination diet
that should only be done from 2-6 weeks. After which, you are encouraged to slowly
reintroduce FODMAP-rich foods back into your diet. The idea behind the short-term
elimination of FODMAP-rich foods from patients' diets is to figure out exactly which
FODMAP-rich foods are causing digestive discomfort. Once patients have identified which
particular FODMAP-rich foods are triggering their symptoms, they are then in a better
position to adjust their dietary plans and minimise the occurrence of bothersome
symptoms.
I agree with the comment, that an elimination phase should be done for 2-6 weeks after which
patients are encouraged to slowly reintroduce FODMAP-rich foods back into their diet. That
is what we have recommended to our patients. The study included a consultation with a
dietician after 6-week elimination diet. Patients were advised to reintroduce eliminated food
products, one after another, with careful observation of the body’s reaction to the introduced
product. However, in our study, we examined the effect after the elimination phase of the
low-FODMAP diet. We added that information in lines 118-122.
11. The discussion of study limitations was incomplete. What about recall issues/bias?
To limit recall bias the patients were advised keep food diaries. They were supposed to be
completed immediately after meal consumption. Patients presented diaries to a dietitian on
follow-up visit. However, its self-reported nature might have influenced the compliance
(lines: 115-118 and 314-316).
12. Please have a separate conclusions section.
We separated conclusions section (line 331) as advised.
13. Please include a data availability statement. The underlying data should be made
publicly available. If this was not possible, please provide a reason why.
The information is provided in lines 357-358.

Reviewer 2 Report

The aim of the study was to evaluate the effectiveness of a low-FODMAP diet in IBS patients whilst in IBD remission.

A low FODMAP diet restricts high FODMAP foods. Scientific evidence suggests that this eating pattern may benefit people with IBS. May reduce digestive symptoms. The disadvantage of this diet is that it excludes many healthy foods such as legumes, many types of fruit and vegetables and some types of cereals. 

The study is conducted quite well but There are a few issues to be resolved.

1. There is some confusion as to which opazients were included in the study. IBS at what stage? 

2. As mentioned, the pros and cons of the FODMAP diet must be clarified

3. The discussion is very sterile. The authors need to investigate the possible clinical usefulness of the FODMAP diet in these subjects and what the risks of poor nutrition are in the long term. The data also highlight the uselessness of the SIBO test. Questio tes are widely misused by many nutritionists. The uselessness of this type of test, particularly in IBS patients, should be emphasised in the discussion.

4. There is little nutrition in this study. 

5. Be careful because there are some plagiarised paragraphs mainly from here https://pubmed.ncbi.nlm.nih.gov/34836367/

Author Response

Thank you for your comments and suggestions to the manuscript. We would be very grateful
to reply as detailed as possible.
1. There is some confusion as to which of patients were included in the study. IBS at what
stage?
As suggested, we specified information on subtypes of IBS diagnosed in patients in lines 141-
144. 7 patients had IBS with predominant constipation (IBS-C), 26 had IBS with predominant
diarrhoea (IBS-D), 1 patient suffered from IBS with mixed bowel habits (IBS-M), 31
participants were diagnosed with unclassified IBS (IBS-U).
2. As mentioned, the pros and cons of the FODMAP diet must be clarified.
We added the pros and cons of the low-FODMAP diet to the introduction section (lines 59-
63).
3. The discussion is very sterile. The authors need to investigate the possible clinical
usefulness of the FODMAP diet in these subjects and what the risks of poor nutrition are in
the long term. The data also highlight the uselessness of the SIBO test. They are widely
misused by many nutritionists. The uselessness of this type of test, particularly in IBS
patients, should be emphasised in the discussion.
Thank you for the suggestion. We have added additional information on the possible
usefulness of the low-FODMAP diet and risks of malnutrition in lines 264-273. Moreover, we
suggested a more thoughtful approach to LHBT performance in lines 306-311.
4. There is little nutrition in this study.
I am not certain if I understand the comment properly. We added additional information and
details on low-FODMAP dietary recommendations in lines 107-114 and 118-122.
5. Be careful because there are some plagiarised paragraphs mainly from here
https://pubmed.ncbi.nlm.nih.gov/34836367/
Więcek and Szczubełek is the same person (surname changed after marriage). We changed
some sentences to avoid self-plagiarism. 

Reviewer 3 Report

This study is a non-randomized controlled trial which evaluates the effectiveness of a low-FODMAP diet in IBD patients who meet criteria for IBS. The results of the study are interesting and pave the way for further research. However there are methodological issues that should be resolved or discussed in the limitations of the study.

1. The authors state in the discussion of the manuscript that compliance of patients with FODMAP diet was self-reported by the participants. This is one of the major limitations of the study, as it is highly unlikely that almost 100% (96.6%) of patients were compliant with this very restrictive diet. Please, provide more details in the method section on how exactly the compliance with the diet was evaluated, by whom, and if any of available dietary methods (24h-dietary recall, 3- or 7-day food record, etc.) were used. If this data is not available, please state this in the limitations of the study.

2. In methods, I suggest to the authors to add a short paragraph regarding dietary exclusions on low-FODMAP diet that was prescribed to the patients. 

3. Since there is no scientific consensus on the positive SIBO test, can authors please provide more details on how LHBT was performed and how were the results interpreted? What was considered a positive test?

4. How were IBS symptoms (flatulance, diarrhea, constipation) determined? Subjectively by patients or objectively using questionnaires or some other method? Please explain in the method section.

5. Was the quality of life of patients determine, and if not, please elaborate. 

6. Authors have stated that all parameters were determined at the start and after 6 weeks of FODMAP. Therefore, please add in the results (for example in Table 2) anthropometric measures (body weight, height and BMI) at the start and at the follow-up, as this is very important data, especially for IBD patients on a restrictive diet.

7. I suggest to the authors to test for the difference in effectiveness of FODMAP diet between CD and UC patients. 

8. Please change the last sentence in the Abstract, as it is not possible to conclude this based on the evidence provided by the study.

Author Response

(The authors gave the same response as above.)

Round 2

Reviewer 1 Report

Thank you for the revisions. Some further comments for the authors to address, please.

Specific comments:

1. Manuscript is still in general need of wordsmithing. For example, please change "Low-FODMAP intervention can lead ..." to "Low-FODMAP intervention is not meant for long-term use as it can lead ...".

2. In terms of study limitations, it should also be mentioned that information on nutritional supplements and complementary and alternative therapies use, which are popular among these patients, was collected. Supplements and vitamins such as Vitamin D may have beneficial effects on IBS symptoms (citation: pubmed.ncbi.nlm.nih.gov/35396764).

3. Compliance in this study may have been high also because of the relatively small and screened/selected group of participants.

4. "... improved the intensity of gut symptoms" - do you mean 'reduced the severity of gut symptoms'?

5. Given the quality of evidence in the present study and in general, I would actually caution authors against making the conclusion that a low FODMAP diet "can therefore be recommended without an LHBT". Moreover, key practice guidelines e.g. those by the American College of Gastroenterology only offer a conditional recommendation of a limited trial of a low FODMAP diet in patients with IBS to improve global IBS symptoms, based on the very low quality of evidence overall.

Author Response

Thank you for your comments and suggestions to the manuscript. We would be very grateful
to reply as detailed as possible.
1. Suggest including the definition of IBS in the introduction section. It is a combination of
chronic abdominal pain associated with a change in the frequency or form of stool.
We included the definition of IBS in the introduction section (lines 42-44) as suggested.
2. "... a link between IBS and small intestinal bacterial overgrowth (SIBO)" - apart from this,
chronic, low-grade, subclinical inflammation has been implicated in the disease process and
is thought to perpetuate IBS-like symptoms (citation:
ncbi.nlm.nih.gov/pmc/articles/PMC6159811).
We added that information in lines 44-45.
3. It is unclear how the patients for the study were recruited. Were these patients being
followed up at a tertiary hospital or specialist centre?
200 patients from a gastroenterology outpatient centre (Central Clinical Hospital of the
Ministry of Interior and Administration in Warsaw, Poland) were recruited for the study.
Those with diagnosed IBS were admitted for a one-day hospitalisation in Clinical Department
of Internal Medicine and Gastroenterology with Inflammatory Bowel Disease Subunit, Central
Clinical Hospital of Ministry of the Interior and Administration in Warsaw, Poland to perform
all tests included in the study protocol in week 0 and week 6 of the study. We updated that
information in lines 67-75.
4. Was there an age cut-off for the study participants?
The age cut-off for the study participants was 80 years old. The oldest participant was 79 years
old. We added that information in line 85.
5. It is important to specify if the subjects were IBS-C, IBS-D or IBS-M. This has important
implications for treatment.
We added thisinformation in lines 141-144. 7 patients had IBS with predominant constipation
(IBS-C), 26 had IBS with predominant diarrhoea (IBS-D), 1 patient suffered from IBS with mixed
bowel habits (IBS-M), 31 participants were diagnosed with unclassified IBS (IBS-U).
6. What about subjects taking vitamin supplements? Vitamin D may have beneficial effects
on IBS symptoms (citation: pubmed.ncbi.nlm.nih.gov/35396764). This should be
acknowledged.
We did not collect information on vitamin D supplementation. The patients enrolled in the
study had no modifications in the use of medications and supplements during the last 8 weeks
before the study to avoid its potential impact on the effect of the low-FODMAP diet. We
specified that in lines 87-88.
7. "The two measurements (baseline and follow-up) were compared with a paired t-test or
Wilcoxon signed-rank test and the mean/median differences (MD) with 95% confidence
intervals (CI) were calculated. McNemar’s test was used to compare nominal variables
between baseline and follow-up, while the chi-square test was used to compare the
effectiveness of a low-FODMAP diet between groups" - for a “repeated-measures” scenario
(multiple intervals or considerable matching between groups), you need to reduce the
alpha level using the Bonferroni correction method.
Thank you for this comment. We however believe that Bonferroni correction is not applicable
in this case. Bonferroni correction is used for multiple comparisons of the same group (i.e.
when there are at least 2 comparisons). In our case if we had data from at least 3 time points,
Bonferroni correction would be a good approach (there would be 3 comparisons: 1 vs. 2, 1 vs.
3, 2 vs. 3). In our case we have only 2 time points, so there is only one comparison (baseline
vs. follow-up), so Bonferroni correction for multiple comparisons is not applicable.
8. Please provide the actual institutional review board (IRB) study/approval number for
reference.
The information is provided in lines 349-352.
9. Please change "CI95" to "95% CI".
We changed that as requested.
10. The conceptual basis for a low-FODMAP diet seems to be misunderstood by the
authors. The low-FODMAP diet is a temporary learning diet designed to help ameliorate
digestive symptoms. The foods are not absolute or universal for every patient with IBS;
some patients do fine with oranges while others are triggered by it. It is an elimination diet
that should only be done from 2-6 weeks. After which, you are encouraged to slowly
reintroduce FODMAP-rich foods back into your diet. The idea behind the short-term
elimination of FODMAP-rich foods from patients' diets is to figure out exactly which
FODMAP-rich foods are causing digestive discomfort. Once patients have identified which
particular FODMAP-rich foods are triggering their symptoms, they are then in a better
position to adjust their dietary plans and minimise the occurrence of bothersome
symptoms.
I agree with the comment, that an elimination phase should be done for 2-6 weeks after which
patients are encouraged to slowly reintroduce FODMAP-rich foods back into their diet. That
is what we have recommended to our patients. The study included a consultation with a
dietician after 6-week elimination diet. Patients were advised to reintroduce eliminated food
products, one after another, with careful observation of the body’s reaction to the introduced
product. However, in our study, we examined the effect after the elimination phase of the
low-FODMAP diet. We added that information in lines 118-122.
11. The discussion of study limitations was incomplete. What about recall issues/bias?
To limit recall bias the patients were advised keep food diaries. They were supposed to be
completed immediately after meal consumption. Patients presented diaries to a dietitian on
follow-up visit. However, its self-reported nature might have influenced the compliance
(lines: 115-118 and 314-316).
12. Please have a separate conclusions section.
We separated conclusions section (line 331) as advised.
13. Please include a data availability statement. The underlying data should be made
publicly available. If this was not possible, please provide a reason why.
The information is provided in lines 357-358

Reviewer 2 Report

The authors improved the paper following my suggestions.

It can be published

Author Response

Thank you for your time, substantive tips and comments on manuscript that contributed to increasing the value of our work.

Reviewer 3 Report

The authors have reviewed the paper according to suggestions. However, some methodological questions have risen up:

1) Authors did not mention how long were the food diaries (1,3 or 7 days?).  Also, at what point after FODMAP introduction were food diaries collected? Please add this in Method section.

2) Was there a difference in effectiveness of FODMAP based on type of IBS? Please add this into result section.

3) How was effectiveness of FODMAP defined? Complete resolution of symptoms, diminished symptoms, self-reported by the participant, to whom did they report? This is very important to define well in the Method section.

Author Response

Thank you for your comments. We are very pleased to reply to your suggestions.

1) Authors did not mention how long were the food diaries (1, 3 or 7 days?). Also, at what
point after FODMAP introduction were food diaries collected? Please add this in
Method section.
We collected a 7-day food diary that was prepared by participants a week prior to the
follow-up visit. As suggested, we added that information in Method section in lines 118-
119.
2) Was there a difference in effectiveness of FODMAP based on type of IBS? Please add
this into result section.
We added this information as advised. The effectiveness of low-FODMAP diet in
patients with predominant constipation (IBS-C subgroup) was 71.4% (95%
CI[29.1;96.3]) (5 of 7 patients), in patients with predominant diarrhoea (IBS-D) it was
66.7% (95% CI[44.7;84.4]) (16 of 24 patients), in patients with unclassified IBS (IBS-U) it
was 63.0% (95% CI[42.4;80.6] (17 of 27 patients). No statistically significant difference
in low-FODMAP effectiveness was observed between the IBS-C, IBS-D and IBS-U
subgroups (p>0.999) (lines 209-214). To compare the effectiveness of a low-FODMAP
diet between subgroups Fisher’s exact test was used (lines 137-138).

3) How was effectiveness of FODMAP defined? Complete resolution of symptoms,
diminished symptoms, self-reported by the participant, to whom did they report? This
is very important to define well in the Method section.
Thank you for this question. We consider it very important to supplement this
information in the Methodology section (lines 103-106). Low-FODMAP effectiveness
was defined as complete resolution of abdominal pain and diminished symptoms of
diarrhoea and/or constipation. It was assessed subjectively by the patients and
reported to the physician during the follow-up interview.
Thank you for your time, substantive tips and comments on manuscript that
contributed to increasing the value of our work. 
